# Phytotherapy: A Solution to Decrease Antifungal Resistance in the Dental Field

**DOI:** 10.3390/biom12060789

**Published:** 2022-06-04

**Authors:** Katherine Cuenca-León, Edisson-Mauricio Pacheco-Quito, Yanela Granda-Granda, Eleonor Vélez-León, Aránzazu Zarzuelo-Castañeda

**Affiliations:** 1Academic Unit of Health and Wellness, Faculty of Dentistry, Catholic University of Cuenca, Cuenca 010105, Ecuador; epachecoq@ucacue.edu.ec (E.-M.P.-Q.); kacuenca22@gmail.com (Y.G.-G.); mvelezl@ucacue.edu.ec (E.V.-L.); 2Research Group: Innovation and Pharmaceutical Development in Dentistry Research Group, Faculty of Dentistry, Head of Research and Innovation, Catholic University of Cuenca, Cuenca 010105, Ecuador; 3Pharmaceutical Sciences Department, University of Salamanca, 37007 Salamanca, Spain; drury@usal.es

**Keywords:** antifungal agents, mouth, drug resistance, phytotherapy

## Abstract

The pathologies produced by fungi in the oral cavity in recent decades have become a health problem, with factors such as an imbalance of the local microbiota being the cause for their propagation. Conventional antifungal treatments, instead of being beneficial, have generated alterations that have led to antifungal resistance. The aim of this study was to investigate and describe phytotherapy resources as a possible solution to oral antifungal resistance. A bibliographic search was carried out on platforms such as PubMed, Scopus, ScienceDirect, Web of Science, and Google scholar. A total of 248 scientific articles were obtained, of which 108 met the inclusion criteria. Microorganisms of fungal origin currently show resistance to the different antifungals of conventional use, which is undoubtedly altering the oral health of human beings, but there are new therapeutic possibilities such as the active principles of various natural species.

## 1. Introduction

For a while now, the pathologies derived from the presence of fungi at the oral level have presented a real threat to dentists. Factors such as alterations, changes, or imbalances in the oral microbiota set off the proliferation of these fungi. Another factor that favors the proliferation of fungus is the indiscriminate use of antimicrobials because instead of controlling mycotic disease, these create alterations that benefit the effective decrease and/or loss of the antagonist bacterial flora, bringing the presence of xerostomia in the mouth. Additionally, the use of dental devices, smoking, inadequate nutritional habits, treatments against terminal disease, endocrine disorders, specific immunity variations, a decrease in the antifungal matter of saliva (as histatins), and systematic immunity variations are the factors that, individually or in groups, determine the conditions for the presence of or increase in microorganisms and constitute vectors able to produce tissue damage [1].

Among the fungal pathologies, more common to the buccal level are candidiasis generated by yeast of the genus *Candida*, Zigomicosis (*Mucor* spp., *Rhizopus* spp.)*,* systematic mycosis derived by the dimorphic fungi *Paracoccidioides brasiliensis*, *Histoplasma capsulatum*, *Coccidioides immitis*, and *Blastomyces dermatitidis*, yeasts (*Cryptococcus neoformans*), and *Aspergilosis* (moho *Aspergillus* spp.), with oral candidiasis being the one with highest incidence worldwide [2]. In the therapeutic field, there is a multitude of pharmacological treatments to combat fungi and yeasts in the oral cavity, both topical and systemic, depending on the type and level of infection, among which antifungals and imidazoles (fluconazole, itraconazole, ketoconazole, miconazole, etc.) stand out. Currently, the efficacy of these treatments is reduced because microorganisms have generated resistance to these types of drugs, which is usually associated with the presence of the most resistant yeasts, the appearance of new pathogenic species, the uncontrolled use of antifungals, and the increase in doses [3,4]. In this situation, a therapeutic option of great validity could be the use of various active components that are found in different vegetable species, which is an alternative that might decrease both the side effects that are present when using conventional drugs and the resistance to different medicines [5]. Based on this, our aim was to develop a literature review about phytotherapy as an alternative treatment to decrease antifungal resistance in the oral cavity.

## 2. Materials and Methods

For this study, a literature search was performed in different databases such as PubMed, Scopus, ScienceDirect, Web of Science, and Google Scholar. The inclusion criteria were based on the use of keywords such as antifungal agents, mouth, drug resistance, phytotherapy, oral fungi, fungal flora of the oral cavity, and antifungal treatment in the oral cavity. The exclusion criteria were based on discarding articles that did not present the keywords, that were not within the indicated database, by type of study (literature reviews, systematic reviews, meta-analyses, and clinical trials), and articles were not excluded by language. In addition, information referring to the dental field and fungal pathologies was investigated [6]. The information was collected for the time period from 1991 to 2021.

A total of 248 bibliographic sources were obtained, of which 108 met the specific search criteria; this information was processed and analyzed (Figure 1).

## 3. Results

### 3.1. Mycotic and Antifungal Agents in the Oral Cavity

Fungi are part of the oral microbiota, wherein they constitute a minor component; however, if their presence surpasses the balance, they become pathological agents that provide a place to numerous diseases with a significative influence on the population.

It is necessary to point out that progress regarding the use of antifungal drugs has evolved over time (Figure 2) [7]. Table 1 lists the classification of antifungals and their mechanisms of action.

In the oral cavity, when conditions are adversely presented, such as pH imbalance, lack of hygiene, or cases of immunosuppression, several pathologies occur, and even the replication of several opportunistic and pathogenic species such as fungi and bacteria, being the first ones the subject of interest in this review. Fungi in the oral cavity are presented in biofilms of multiple kinds, exhibiting features of an antifungal-resistant phenotype as a result of some factors that change their components and structure. The elimination of these biofilms and their treatments are based on the use of antifungal drugs that allow their treatment and prevention. In addition, antiseptics such as chlorhexidine and triclosan are often used to reduce the microbial load present in the oral cavity, reducing bacteria, viruses, and fungi. However, these therapeutic options in many cases fail to fully improve oral health due to inadequate use and microbial load, enhancing antimicrobial resistance.

### 3.2. Resistance to Antifungals and Their Effect on Human Health

Fungal infections are a public health problem. About 1.7 billion people worldwide suffer from a fungal infection; most of these infections occur at the mucosa or skin level. *Candida* species represent one of the predominant causes of fungal infections with high morbidity and mortality rates [8,9]. *Candida* species are considered one of the most frequently isolated microorganisms in microbiological cultures of fungal-infected patients. The most frequent *Candida* species is *Candida albicans*; however, the incidence of other species such as *Candida tropicalis*, *Candida parapsilosis,* and *Candida glabrata* has been increasing, as well as *Aspergillus* spp. species. This high incidence is caused by the inadequate use of antifungal drugs since the available therapeutic options limit specialized treatment, producing resistance to antifungal drugs. Antifungal resistance generally occurs to echinocadins and azoles, although cross-resistance to two or more antifungal classes, this situation is of concern and requires appropriate countermeasures. Therefore, new strategies are needed to combat this type of pathogens through more potent molecules of plant origin [10].

### 3.3. Drugs That Produce Antifungal Resistance

Treatment of mucosal or invasive candidiasis is based on a set of limited antifungal agents, which include polyenes, azoles, and echinocandins [11].

Amphotericin B, an antifungal drug widely used in the 1980s, is toxic due to its lack of selectivity; since fungal cells are eukaryotic, it is currently only used for the treatment of serious fungal infections that threaten life. This fact has led to the search for new active principles, highlighting the azoles, which lack toxicity but are fungistatic drugs, which degenerate into resistance.

In 2001, a new class of antifungals came onto the market, the echinocandins (caspofungin), considered a broad-spectrum antifungal and effective against infections caused by species of *Candida* spp. or *Aspergillus* section *Fumigatti.* Unfortunately, their clinical use is limited to the treatment of systemic candidiasis and causes the development of resistance, especially to *C. glabrata*, in some cases. According to investigations, resistance occurs in between 2.9% and 3.1% of cases [12], typically corresponding to acquired resistance after exposure to echinocandins [13,14].

The effect of nystatin, an antifungal from the polyene group that is able to act as a fungistatic or fungicide, depends on the concentration. An example of this effect is on the ergosterols of a fungus’ cell membrane, where it shows greater affinity, modifying it spatially, affectation of the cellular metabolism, and also has an affinity for cholesterol, making it a drug with toxic potential limiting its use [15].

Fluconazole, on the contrary, has limitations due to the natural resistance of some fungi toward the compound (*Candida krusei*), the power of some species of *Candida* to achieve resistance to established doses, along with the necessary adjustments in drugs, in renal patients, and the mutual action that can be exerted with another medication. Following this line, evidence indicates that certain species of *Candida* tend to develop resistance to fluconazole by the influence of intrinsic factors such as pathogenicity.

### 3.4. Resistance Mechanisms to Anti-Fungal

Resistance mechanisms can be primary or secondary, depending on the intrinsic characteristics of each fungus [16]. The sequential administration of specific drugs generates the appearance of isolates resistant to specific molecules, proving—by in vivo assays—the rapid appearance of mutants resistant to multiple drugs under the condition of combined therapy, a precedent that forces to limit therapy only to exceptional situations [17]. At present, there is a notable decrease in the effect of the drugs used to combat fungi in the oral cavity due to the resistance of microorganisms to the drugs used, mainly because of the emergence of resistant yeasts, the appearance of new harmful species, over-prescription, and the increase in therapeutic doses, as mentioned in previous lines [10]. Species such as *Candida* are competent in understanding and preventing the long-term advantage of phenotypes related to virulence and drug resistance. On the contrary, the authors such as Costa C. et al. and Cuenca M. et al. define the term drug resistance, in this case regarding antifungals, as the simultaneous acquisition of tolerance to several drugs due to a genetic change, where the pathogenic microorganism is inhibited by a higher than normal concentration of the drug, observed in common strains [18,19].

Dental prostheses are a risk factor for subprosthetic candidiasis and can cause mortality in hospitalized older adults. Fungal resistance is mainly due to the ability to produce biofilms, and this can often lead to dental therapeutic failure and requires the investigation of alternatives for treatment [20]. 

A study by Cruz-Quintana et al. explained that it is important to carry out studies on the genome of oral fungi in order to determine the role played by genes as potential therapeutics, but it is necessary to deepen studies on the molecular mechanisms that produce drug resistance by these fungi. Thus, it was determined that the genome of *C. albicans* is of the utmost importance since it will allow the establishment of safe diagnostic protocols in the future, as well as the development of new antifungals that allow its control [21]. In the face of antifungals, it is mandatory to identify, with microbiological techniques based on various identification criteria, several species [22].

In 1994, a biochemical method was described based on revealing species-specific enzymatic alterations through hydrolysis using a substance with chromogenic attributes to reveal GCL (crevicular gingival fluid) based on a glow of colors that occurs when they develop colonies. Without a doubt, this will allow generating more effective treatments, as well as finding alternatives with greater viability to control the pathologies that significantly affect oral health [23].

When talking about resistance and its mechanisms, we can mention those generated against echinocandins and azoles, determining the need to investigate pharmacological options against the resistance generated toward traditional drugs [11].

### 3.5. Phytotherapy Alternatives to Address the Resistance of Antifungals in the Oral Cavity

Traditional herbal medicine known as phytotherapy currently occupies a very important place as a therapeutic alternative, although it is true that a disadvantage is the lack of phytotherapeutic evidence in some species, which does not allow to give the necessary endorsement for its use. However, it is necessary to pave the way for the discovery of new drugs following the ethnobotanical clues of medicinal plants of conventional use [24].

Martínez et al. state that in the presence of a problematic derivative of resistance to antifungal treatments in the dental field, plenty of investigations have been conducted in recent years, with the goal of controlling these pathologies and their effects on the health of the oral cavity. An alternative is the use of phytotherapy through indigenous or native plants [25]. A previous study on phytotherapy showed the in vitro antifungal effect of different hydroalcoholic concentrations of Uncaria Tomentosa (UT) on *C. albicans* (ATCC), highlighting that a hydroalcoholic extract of UT at 100%, such as nystatin, is more effective on fungi species because of its high sensitivity; however, concentrations of 50% and 75% result in a medium sensitivity, while resistance can be observed at a 25% concentration [26].

It is important to mention that Ayurvedic medicine has long been used to treat dental conditions dating back to 2000 BC; however, phytotherapy in dentistry is not well known and is hardly used. There is evidence of its effects on the antimicrobial, antiplaque, analgesic, healing, antioxidant, and anti-inflammatory spectrum, establishing its viability in the treatment and prevention of dental caries, periodontal disease, oral ulcers, and mucosal wounds [27].

Menezes, A. et al. confirm what was said in the previous paragraph since when evaluating the chemical effects of the essential oil of *Bauhinia rufa flower* as an antifungal against *Candida* spp isolates, it showed an excellent content of essential oil, and the effects derived from its antifungal action place it as a possible alternative to develop a new antifungal; as a result, it obtained a yield of 0. 067%, 0.045%, 0.098%, and 0.065%, with relative densities of 0.907, 0.905, 0.908, and 0.904 g/mL^−1^, respectively, in the study sites, exposing a chemical profile with the presence of 39 mixtures—six with the highest proportions of β-pinene, elemol, globulol, trans-verbenol, viridiflorol, and oplopanone; this shows that the antifungal incidence against *Candida* was excellent in all tests performed [28]. Mansourian A. et al., in their experimental study, identified *C. albicans* in immunosuppressed patients resistant to antifungal drugs, which led these researchers to turn their attention to the medicinal herbs Syzygium aromaticum and Punica granatum. Thus, they worked with 21 oral *C. albicans* isolates from patients with prosthetic stomatitis at the Department of Prosthodontics, Faculty of Dentistry, Tehran University of Medical Sciences; *S. aromaticum* species showed better activity against *Candida* than conventionally used antifungals (*p* < 0.001) [28,29].

In a study by Thamburan et al., patient samples and the standard strain 5027ATCC (PTCC10231) of *C. albicans* yeast were investigated using the well diffusion method. Nystatin and methanol were obtained as positive and negative controls, respectively. The results were conclusive: Both *S. aromaticum* and *P. granatum* showed important antifungal activity using the well method. *S. aromaticum* showed high anti-*Candida* activity in relation to nystatin, generating statistical significance (*p* < 0.001) [30].

Thamburan et al., in their research, also identified a population with clinically significant oropharyngeal candidiasis, which did not allow the provision of oral medications, also hindering food intake. Azole antifungals are generally used in this type of pathology; however, a limitation is the high resistance in the general population and also a number of toxic effects. Therefore, the evaluation of two native South African species called *Tulbaghia alliacea* and *Tulbaghia violacea* was proposed in this research. The researchers compared the in vitro antifungal activity on *Candida* species of natural extracts of *T. alliacea* and *T. violacea*, obtaining important data since it was demonstrated that the extract of *T. alliacea* was a natural fungicide. This activity could be due to an active component called marasmicin, concluding that the extracts of *T. alliacea* showed anti-infective activity against *Candida* species in vitro [30].

All of this shows that studies related to phytotherapy provide favorable results in terms of antifungal treatments in the oral cavity against the resistance generated by fungi according to conventional pharmacological treatments. Even in the case of hepatotoxicity generated by antifungal drugs, phytotherapy shows an outstanding advantage. The following is a compilation of several studies on plant alternatives to overcome drug resistance in the oral cavity as a therapeutic option, describing the plant family, plant common name, botanical name, bioactive compounds and type of extract, fungus on which it acts, and main findings (Table 2).

As shown in Table 2, essential oils derived from different plant species are used for various pathologies, including some infectious diseases, so the antibiofilm activity of their active components continues to be the subject of recent research. There are studies showing that there are plant species that have been little investigated and whose antifungal activity is known, such as *T. vulgaris* and thymol, whose function is based on the eradication of *Candida* species and which have shown resistance to drugs, especially *C. tropicalis* [86].

The limited access to antifungal drugs for the specific treatment of *C. albicans* in the oral cavity has promoted research on products of natural origin for the discovery of new therapeutic possibilities. Based on this context, tropical countries are pioneers in the production of these natural products with potential antimicrobial activity. Walicyranison Plinio Silva-Rocha et al., in their research on the effect of *E. uniflorain* on *C. albicans*, found that more than 80% of A549 cells (human alveolar epithelial cell line) remained viable even when exposed to four times the concentration of *E. uniflora* EC (8000 μg/mL; unpublished data), and *E. uniflora* EC inhibited hyphal formation in both the liquid and solid media tested. It also impaired hydrolytic enzyme production. This is one of the first studies to describe the interaction of a natural product with the full expression of three different factors in *C. albicans.* Therefore, *E. uniflora* may be a therapeutic alternative for oral candidiasis in the future. Thus, phytotherapy could become a future option, as previous studies have shown [87].

In some parts of the world, such as in the city of Blumenau, several practices such as acupuncture, homeopathy, phytotherapy, hydrotherapy, and anthroposophical medicine are offered by health professionals, including doctors, nurses, dentists, oral health technicians, and nurses. In the study proposed by Mattos et al., it was reported that most health professionals (96.2%) know and believe in the therapeutic effects of medicinal plants but do not prescribe them in their practice. A large percentage of professionals (98.7%) believe that it is a priority to include this practice to complement health services, with the aim of reducing the resistance and side effects produced by the drugs commonly used. These investigations show the importance of the use of phytotherapy; the idea is to advance studies and standardize protocols for the use and management of these active components through experimental studies [88].

### 3.6. Discussion

The oral cavity is the entrance of viruses, bacteria, and fungi from the surrounding environment, becoming the habitat with the highest microbiological population density, referring to the human body. The oral cavity constitutes a very important niche for housing various types of microorganisms, which together form the biofilm. The mouth houses approximately six million bacteria and around 35 times more viruses, as well as an undetermined number of fungi [2].

The presence of biofilm causes various oral pathologies, such as dental caries, gingivitis, periodontitis, peri-implant mucositis, peri-implantitis, or fungal infection processes such as oral candidiasis. In this situation, prophylactic and conventional treatments seek to reduce the microbial load at the oral level and thus reduce the possible pathologies that may occur. However, when there is a microbial imbalance or microbial alteration, the presence of opportunistic infections is very frequent, which often leads to the use of conventional drugs to reduce pathologies. In this situation, prophylactic and conventional treatments seek to reduce the microbial load at the oral level and thus reduce the possible pathologies that may occur. It is here where the dentist plays a crucial role, trying to promote proper dental health or, in severe cases, trying to alleviate the disease.

In this article, we attempt to elucidate this problem, considering studies focused on the use of antifungal drugs such as nystatin, β-amphotericin, azoles, polyenes, and echinocandins, and their resistance—in some cases specifically and in others more generally—against the control of oral cavity fungi.

Thus, several authors such as the Spanish Association of Pediatric Dentistry and Cruz et al. [89] demonstrated the inappropriate use of drugs during childhood is one of the reasons for the higher incidence of resistance to antifungals. This situation occurs when the dentist who has prescribed treatment does not have adequate and sufficient information about the drug, specifically about therapeutic indications, pharmacokinetics, pharmacodynamics, drug–drug interactions, contraindications, dosage, and adverse effects.

Thus, the lack of reliable clinical information and diagnostic tests further exacerbates the problem. To this must be added the errors made in the dosage and administration of medications to elderly patients, not to mention the free and indiscriminate access to certain drugs.

Undoubtedly, and as stated by Menezes et al. [28], many well-known and over-the-counter antifungal drugs currently maintain low fungistatic activity due to the presence of resistance, mainly of the *Candida* genus, which is considered to be the most prevalent in the oral cavity. Within the same context, Peres et al. stated that, in general, the control of fungal infections depends mainly on the host’s immune response. The pathology manifests itself when there is a failure in the defenses or the pathogen eludes the responses, which undoubtedly leads to the use of fungicidal or fungistatic drugs that intervene specifically on the aggressor agent, as opposed to the possible damage caused to the host. In addition, the usual antifungal agents maintain a limited number of cellular targets, such as ergosterol and the enzymes involved in its synthesis, nucleic acid synthesis, and cell wall [90].

On the other hand, Pontón and Quindós declared resistance to antifungals is associated with the morbidity and mortality of mycoses; clinical outcomes in the face of antifungal resistance can be seen in procedural failures and mutations of fungal agents.

Another situation that causes resistance to antifungals would be the limitation of existing pharmacological treatments. This is mentioned by Donnelly et al. and Bulacio et al.; in their studies, they determined that some types of *Candida* were sensitive to clotrimazole and nystatin, drugs generally used topically; however, in the case of miconazole, this was not the case. Therefore, the level of difficulty in the treatment of these infections is mainly due to the limited number of antifungal drugs available, the presence of increasingly resistant strains, and the ability of these microorganisms to generate biofilms [91,92,93].

Fungi assembled in biofilms have a unique ability that is approximately 1000 times greater than planktonic cells to resist antifungal drugs, even without carrying explicit resistance genes. It is important to mention that antifungal treatment, compared to antibacterial therapy, is limited in terms of drugs since the treatment is long and its side effects are serious, as well as there being very frequent drug interactions. Another situation to be considered is the one mentioned by Rivera-Toledo et al., who stated that infections by fungal microorganisms in immunocompetent people are increasing, probably due to the incorporation of more virulent species and variants resistant to antimycotic agents or due to the introduction and adaptation of fungi to new ecological niches, as is the case of *Cryptococcus gattii* in Canada [94].

Lastly, it is worth mentioning other risk factors described by Rey et al.; they mentioned established that the colonization of the mouth by fungi is very common in healthy people, and its presence is more common in older adults (7–65%). Among the elements that affect and produce this type of condition are sex, salivary alterations (quantitative and qualitative), use of prostheses, smoking, and state of health (mainly immunological or endocrine changes, prolonged pharmacological treatments, etc.). As has been observed throughout this review, antifungal resistance is increasing, which allows us to affirm that it is of utmost importance to investigate alternative antifungal strategies capable of controlling these pathologies that affect world health [13].

The response to the problem of resistance to antifungal drugs and their various side effects has not been long in coming. Therapeutic alternatives for the treatment of oral cavity fungi are being widely investigated and present some possibilities, from new and more potent chemical molecules to phytotherapy [95,96,97,98].

Phytotherapy is an option against this problem, as it can be an alternative to current treatments for oral health problems since there are several studies that have shown that these natural herbal treatments have fewer side effects and provide greater safety versus conventional treatments [5].

In a study conducted in Mexico by Cruz et al. in which they investigated several medicinal plants with antimicrobial properties used in the oral setting based on ancient medicine books and healers, they determined that they could serve for therapeutic purposes for periodontal disease or as anti-cariogenic agents. However, a limiting factor is the shortage of preliminary studies that support their use [25].

As far as is known, there is little information on the use of plant extracts tested in clinical and animal trials. Varoni et al., in their research, compiled data on this topic from 1965 to 2011 and reported results on grapes, berries, tea, cocoa, coffee, myrtle, chamomile, honey/propolis, aloe vera, and three groups of phenols (i.e., flavonoids, stilbenes, and proanthocyanidins), investigating their effect on oral pathologies such as gingivitis, caries, periodontal disease, candidiasis, oral thrush, oral mucositis, oral lichen planus, leukoplakia, and oral cancer. Preclinical studies have suggested interesting activities of these three polyphenols on more recurrent oral diseases (candidiasis, caries, and periodontitis) [99]. For this reason, in their research work, they focused their interest on the plant kingdom, which offers several advantages for various pathologies in the dental field, especially active phytocompounds, emphasizing their chemical structure and their mechanisms of action. The authors mentioned that plants are a promising therapy compared to conventional therapy ones [100].

Finally, it is important to note that in some parts of the world, such as in the city of Blumenau, several practices such as acupuncture, homeopathy, phytotherapy, hydrotherapy, and anthroposophical medicine are offered by health professionals, including doctors and dentists. In the study proposed by Mattos et al., it was reported that most health professionals (96.2%) know and believe in the therapeutic effects of medicinal plants but do not prescribe them in their practice. A large percentage of professionals (98.7%) believe that it is a priority to include this practice to complement health services, with the aim of reducing the resistance and side effects produced by the drugs. These investigations show the importance of the use of phytotherapy; the idea is to advance studies and standardize protocols for the use and management of these active components through experimental studies [88,101,102,103].

In this review, we focused specifically on phytotherapy as a support option to conventional treatment. It is necessary to mention that although there have been several research works, it is important to go deeper into the subject in order to determine the active compounds and their efficacy against typical pharmacological treatments. Another important factor is the planning and execution of regulated and standardized protocols for the practice, with respect to the type of active component, concentration, and period of use [102,103].

## 4. Conclusions

Conventional antifungal treatments in dentistry, such as nystatin, amphotericin β, azoles, corticoids, polyenes, and echinocandins, are currently highly resistant. Factors such as prevalence, reduced number of antifungals, the appearance of increasingly resistant strains, their capacity to develop biofilms, prolonged treatments, and poor administration of drugs by specialists and patients are some of the causes of the persistence of the problem. All of this denotes the need to present strategies that allow us to investigate and formulate new therapeutic options based on natural active components through experimental studies, generating research from our reality, reducing drug toxicity and re-resistance, and thus improving the quality of life of our patients, most of whom are immunocompromised.

## Figures and Tables

**Figure 1 biomolecules-12-00789-f001:**
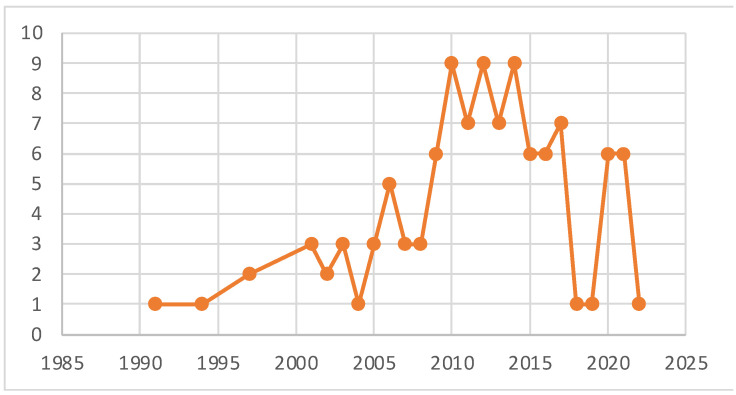
Analyzed articles in this study by year of publication.

**Figure 2 biomolecules-12-00789-f002:**
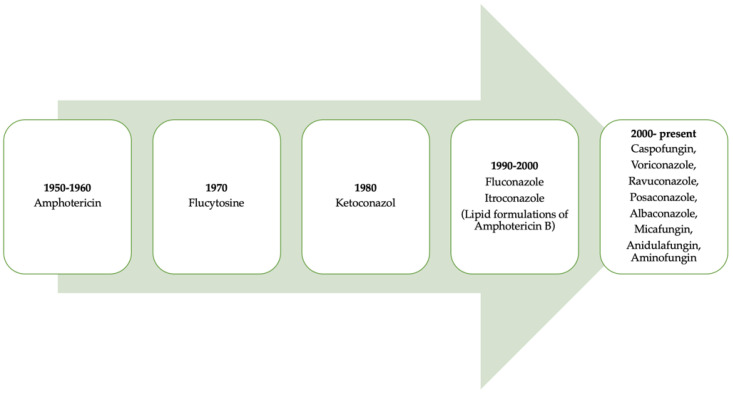
A retrospective look at the evolution of antifungals throughout history. Figure adapted from [7].

**Table 1 biomolecules-12-00789-t001:** Antifungal classification according to structure and mechanism of action [7].

	Inside the Structure	Mechanism of Action
Polyene	Antifungal classification: a look inside the structure Nystatin, natamycin, and amphotericin B	Antifungals that act on the cytoplasmatic membrane
Azoles	Imidazole: miconazole, clotrimazole, and ketoconazole Triazoles: fluconazole and itraconazole (ketoconazole)	Antifungals that act on the cytoplasmatic membrane
Allylamines	Terbinafine and naftifine	Antifungals that act on the cytoplasmatic membrane
Lipopeptides	Papulacandins and glycosylated triterpenes echinocandins: caspofungin, anidulafungin, and micafungin	Antifungals that act on the cell wall
Pyrimidines	Flucytosine	Antifungals that act on the cell nucleus
Other	Ciclopirox potassium iodide, tolnaftate, and griseofulvin	Antifungals that act on the cell nucleus

**Table 2 biomolecules-12-00789-t002:** Plant species with antimycotic activity for the different species of *C. albicans*.

Plant Family	Plant Common Name	*Botanical Name*	Type of Extract	Fungus on Which It Acts	Active Components	Reference
Anacardiaceae	Marula	*S. birrea*	Aqueous extracts	*C. parapsilosis*	Tannic acid, quercetin, phenols, flavonoid, and flavonols	[31]
Annonaceae	Candida lusitaniae	*X. aethiopica*	Aqueous extracts	*C. albicans*	B-pinene, terpinen-4-ol, sabinene, a-phellandrene, a-terpineol, and trans-b-ocimene	[32]
				*C. glabrata*		
				*C. guilliermondii*		
				*C. krusei*		
				*C. parapsilosis*		
				*C. tropicalis*		
				*C. albicans*		
	False nutmeg or calabash nutmeg	*M. myristica*	Aqueous extracts	*C. krusei*	a-phellandrene, p-cymene, a-pinene, cis- sabinol, and limonene	[32]
	Sugar apple or custard apple	*Z. leprieurii*	Aqueous extracts	*C. albicans*	Trans-b-ocimene, a-terpinolene, 3-d-carene, limonene, myrcene, a-pinene, and p-cymene	[32]
	Ethiopian pepper	*Z. xanthoxyloıdes*	Aqueous extracts	*C. albicans*	a-pinene, trans-b-ocimene, citronellol, sabinene, myrcene, limonene, and cytronellyl acetate, a- phellandrene	[32]
				*C. krusei*		
				*C. parapsilosis*		
				*C. tropicalis*		
				*C. albicans*		
				*C. krusei*		
				*C. parapsilosis*		
				*C. tropicalis*		
				*C. albicans*		
				*C. krusei*		
				*C. parapsilosis*		
				*C. tropicalis*		
Anisophylleaceae	Monkey apple	*A. laurina*	Aqueous extracts	*C. albicans*	Flavonoids, phenolics, citric acid, malic acid, tartaric acid, fumaric acids, oxalates, phytic acid, and tannins	[33]
Acanthaceae	False waterwillow	*A. paniculataa*	Methanolic extracts	*C. krusei C. albicans C. tropicalis*	3-O-β-d-glucosyl-14-deoxyandrographiside, 14-deoxyandrographolide,14-deoxy-11,12-didehydroandrographolide, and 14-deoxyandrographolide	[34]
Acoraceae	Sweet flag or calamus	*A. calamus*	Aqueous extracts	*C. albicans C. krusei C. lusitaniae C. parapsilosis*	Triploid and tetraploid flavonoids and lectins	[35]
Amaryllidaceae	Garlic	*A. sativum*	Alcoholic extracts	*C. albicans*	Not reported	[36]
	Onion	*A. cepa*	Alcoholic extracts	*C. albicans*	Tannins and flavonoids such as quercetin	[36,37]
Apiaceae	Coriander	*C. sativum*	Aqueous and alcoholic extracts	*C. albicans*	Linalol, 1-decanol, 2e-decenol, 2z-dodecenol, aldehydes, and 3-hexenol	[38,39]
				*atcc 90028*		
	Cumin	*C. cyminum*	Aqueous extracts	*C. albicans*	Pinene, cineol, and linalool	[40]
	Fennel	*F. vulgare*	Aqueous extracts	*C. albicans*	Trans-anethol, limonene, and fenchone	[41]
	Persian hogweed	*H. persicum*	Hydroalcoholic Extracts	*C. albicans*	Anethol and terpinolene	[42]
	Anise	*P. anisum*	Alcoholic extracts	*C. albicans*	Anethol and coumarins	[43]
Apocynaceae	White’s ginger	*M. whitei*	Hydroalcoholic Extracts	*C. guilliermondii*	Alkaloids, anthocyanins, anthraquinones, flavonoids, phenols, and saponins	[44]
				*C. albicans*		
				*C. lusitaniae*		
				*C. tropicalis*		
Arecaceae	Silk rubber	*Funtumia elastic*	Aqueous and ethanol extracts	*C. albicans*	Anthocyanins, butacyanin, flavonoids, steroids, and tannins	[45]
Asteraceae	Dhangri bet or rab bet	*C. leptospadix*	Ethanol extracts	*C. albicans*	Ursolic acid (triterpenoid saponin)	[46]
	Wormwood	*A. sieberi*	Aqueous extracts	*C. glabrata*	Β-thujone, camphor, and α-thujone	[47]
	Wild rhubarb or lesser burdock	*A. minus*	Ethanol extract	*C. albicans* *C. dubliniensis* *C. glabrata* *C. krusei* *C. stellatoidea C. tropicalis*	Flavonoids (isoquercitrin and rutin), and five minor flavonoids (astragali, kaempferol 3-o-rhamnoglucoside, quercetin 7-o-glucoside, an isomer of quercitrin, and quercetin 3-o-arabinoside), and arctiine	[48,49]
	Field wormwood	*A. campestris*	Aqueous extracts	*C. glabrata* *C. lusitaniae* *C. tropicalis* *C. krusei* *C. parapsilosis*	Triploid and tetraploid flavonoids and lectins	[35]
	Fringed sagebrush or pasture sage	*A. frigida*	Aqueous extracts	*C. parapsilosis* *C. lusitaniae* *C. krusei* *C. tropicalis* *C. glabrata*	Triploid and tetraploid flavonoids and lectins	[35]
	Tall goldenrod or giant goldenrod	*S. Gigantea*	Aqueous extracts	*C. tropicalis*	Triploid and tetraploid flavonoids and lectins	[35]
				*C. lusitaniae*		
				*C. albicans*		
				*C. krusei*		
				*C. glabrata*		
	Yarrow	*A. biebersteinii*	Aqueous extracts	*C. albicans*	Limonene	[50]
Betulaceae	Green alder	*A. viridis*	Aqueous extracts	*C. albicans*	Triploid and tetraploid flavonoids and lectins	[35]
				*C. glabrata*		
				*C. parapsilosis*		
				*C. krusei*		
				*C. lusitaniae*		
	Yellow birch	*B. alleghaniensis*	Aqueous extracts	*C. parapsilosis*	Scalene triterpene and tetracosan aliphatic hydrocarbon	[35]
				*C. albicans*	Triploid and tetraploid flavonoids and lectins	
				*C. krusei*		
				*C. lusitaniae*		
				*C. glabrata*		
Bignoniaceae	Golden bellbean	*M. obtusifolia*	Ethanol extracts	*C. albicans*	Ursolic acid, pomolic acid, and 2-epi-tormentic acid	[51]
	Flame vine	*P. venusta*	Ethanolic extracts	*C. krusei*	Isoverbascoside, verbascoside, and quercetin3-o-x-l rhamnopyranosyl-(1-6)-b-d-galactopiranoside	[52]
				*atcc 6258*		
				*C. krusei*		
				*usp 2223*		
				*C. albicans*		
				*atcc 10231*		
				*C. albicans* *usp*		
				*C. albicans*		
				*C. parapsilosis usp 1933*		
				*C. tropicalis * *usp*		
				*C. guilliermondii usp 2234*		
	Cricket vine	*A. chica cerrado*	Acetone extracts	*C. glabrata*	Phenolics, flavonoids, anthocyanins, β-carotene, and lycopene	[53]
				*C. rugosa*		
				*C. albicans*		
Caesalpiniaceae	Pink trumpet tree or lavender trumpet tree	*T. avellanedae*	Hydroalcoholic extracts	*C. albicans*	Naphthoquinones based on the naphtho [2,3-b]furan-4,9-dione skeleton such as (-)-5-hydroxy-2-(1′-hydoxyethyl)naphtho [2,3-b]furan-4,9-dione	[54]
	Divida	*S. zenkeri*	Hydroethanolic extracts	*C. guilliermondii*	2,4,5,7-tetrathiaoctane	[55]
				*C. parapsilosis*		
				*C. tropicalis*		
				*C. glabrata*		
				*C. krusei*		
				*C. lusitaniae*		
				*C. albicans*		
Caricaceae	Papaya	*C. papaya*	Aqueous extracts	*C. albicans*	N-acetyl-beta-D-glucosaminidase	[56,57]
Combretaceae	Tanibuca	*B. tomentosa*	Methanol extracts	*C. albicans*	Gallic acid	[58]
				*C. tropicalis*		
				*C. krusei*		
				*C. glabrata*		
				*C. parapsilosis*		
				*C. dubliniensis*		
	Bushwillow	*C. albopunctatum*	Hydroalcoholic extracts	*C. albicans*	Terpenoids, flavonoids, phenanthrenes, and stilbenoids Pentacyclic triterpenes, hydroxymberbic acid 2500, and arjunolic acid	[59,60,61]
		*C. imberbe*				
		*C. nelsonii*				
Curtisiaceae	Assegai tree	*C. dentata*	Hydroalcoholic extracts	*C. albicans*	Phenols, flavonoids, tannic acid, saponins, steroids, and alkaloids	[62,63]
Cucurbitaceae	Bitter apple or bitter cucumber	*C. colocynthis*	Acetone extracts	*C. albicans*	Glycosides and resins	[64]
				*C. glabrata*		
				*C. krusei*		
				*C. parapsilosis*		
				*C. guilliermondii*		
				*Candia tropicalis*		
				*C. dubliniensis*		
Ebenaceae	Gabon ebony	*D. crassiflora*	Hydroalcoholic extracts	*C. glabrata*	Isoarborinol methyl ether (cylindrine)	[65]
				*C. albicans*		
				*C. krusei*		
				*C. tropicalis*		
	Evergreen tree	*D. canaliculata*	Hydroalcoholic extracts	*C. albicans*	Plumbagin and two known pentacyclic triterpenes (lupeol and lupenone)	[65]
				*C. kefyr*		
				*C. parapsilosis*		
Eriocaulaceae	Leiothrix	*L. spiralis*	Methanolic extract	*C. albicans*	flavonoids luteolin-6-C-β-D-glucopyranoside, 7-methoxyluteolin-6-C-β-D-glucopyranoside, 7-methoxyluteolin-8-C-β-D-glucopyranoside, 4′-methoxyluteolin-6-C-β-D-glucopyranoside, and 6-hydroxy-7-methoxyluteolin and the xanthones 8-carboxymethyl-1,5,6-trihydroxy-3-methoxyxanthone, 8-carboxy-methyl-1,3,5,6-tetrahydroxyxanthone	[66]
				*C. krusei*		
				*C. parapsilosis*		
				*Candia tropicalis*		
Euphorbiaceae	Pillpod sandmat	*E. hirta*	Hydroalcoholic extracts	*C. albicans*	Β-amirine and 24-methylenecycloarthenol	[67]
	Red sacaca	*C. cajucara*	Methanolic extracts	*C. albicans * *atcc 51501*	Linalool	[68]
	Prostrate spurge or blue weed	*E. prostrata*	Methanolic extracts	*C. albicans*	Flavonoids such as apygenin-7-glycoside, luteolin-7-glycoside, and quercetin phenolic compounds such as ellagic acid, gallic acid, and tannin	[69]
Fabaceae	Prekese	*T. tetraptera*	Hydroalcoholic extracts	*C. glabrata*	Oleanolic glycosides and cinnamic acids	[70]
				*C. krusei*		
				*C. tropicalis*		
				*C. albicans*		
				*C. guilliermondii*		
				*Candia lusitaniae*		
				*C. parapsilosis*		
	Red propolis	*D. ecastaphyllum*	Hydroalcoholic extracts	*C. albicans*	Formononetin	[71]
				*atcc 76645*		
				*C. albicans*		
				*lmp-20*		
				*C. tropicalis*		
	Naranjito	*S. simplex*	Hydroalcoholic extracts	*C. albicans*	Diterpenes	[72]
	Golden shower tree	*C. fistula*	Hydroalcoholic extracts	*C. albicans*	Cassico acid (rhein) and other phenolic compounds	[73,74]
				*C. glabrata*		
				*C. tropicalis*		
	Licorice	*G. glabra*	Methanolic extracts	*C. albicans*	Formononetin, liquiritigenin, and apigenin	[75]
				*C. glabrata*		
				*C. parapsilosis*		
	Senna	*C. alata*	Methanol extracts	*C. albicans*	Chrysoeriol and stearic acid	[76]
	Salt-tree	*H. halodendron*	Aqueous and ethanolic extracts	*C. albicans*	Salicylic acid, p-hydroxybenzoic acid (ferulic acid), and 4-hydroxy3-methoxy cinnamic acid Phenolic extracts Ferulic and sinapic acids	[77]
Gentianaceae	Common centaury	*C. erythraea*	Ethanolic extracts	*C. albicans*	Secoiridoid glycosides, a group of monoterpenoid compounds, and phenolics (xanthones and flavonoids)	[78]
	Lesser centaury	*C. pulchellum*	Ethanolic extracts	*C. albicans*		
	Spiked centaury	*C. spicatum*	Ethanolic extracts	*C. albicans*		
	Slender centaury	*C. tenuiflorum*	Ethanolic extracts	*C. albicans*		
Lauraceae	Cinnamon	*C. zeylanicum*	Ethanolic extracts	*C. albicans * *atcc 10231*	Cinnamaldehyde, benzaldehyde, and cinnamyl acetate	[79]
				*C. albicans * *atcc 90028 1120*		
				*C. albicans*		
				*C. glabrata*		
				*C. parapsilosis*		
				*C. guilliermondii*		
				*C. krusei*		
				*C. lusitaniae*		
				*C. tropicalis*		
Myrtaceae	Gum coolibah	*E. intertexta*	Methanol extracts	*C. albicans*	1,8-cineole	[80]
	Eucalyptus	*E. globulus*	Hydroalcoholic extracts	*C. albicans*	Cineole, limonene, p-cymene γ-terpinene, α-pinene, and α-phellarene	[81]
	Clove	*S. aromaticum*	Hydroalcoholic extracts	*C. albicans*	Eugenol and thymol	[82]
Piperaceae	Pepper	*P. bredemeyeri*	Hydroalcoholic extracts	*C. albicans*	Trans-β-caryophyllene, caryophyllene oxide, β-pinene, and α-pinene	[83,84]
	Wild pepper	*P. capense*	Ethanolic extracts	*C. albicans*	Trans-β-caryophyllene. In addition, caryophyllene oxide, β-pinene and α-pinene	[84]
				*C. guilliermondii*		
				*C. krusei*		
				*C. parapsilosis*		
				*C. lusitaniae*		
	Black pepper	*P. nigrum*	Bha	*C. albicans*	Trans-β-caryophyllene. In addition, caryophyllene oxide, β-pinene and α-pinene	[84]
			Ethanolic extracts			
	Lacquered pepper	*P. regnellii*	Ethyl acetate	*C. albicans C. krusei*	Ethyl acetate	[84,85]
			Ethanolic extracts			
	West African pepper	*P. guineense*	Ethanolic extracts	*C. parapsilosis*	Beta-caryophylleneTrans-β-caryophyllene. In addition, caryophyllene oxide, β-pinene and α-pinene	[84]
				*C. albicans*		
				*C. glabrata*		
				*C. tropicalis*		
				*C. lusitaniae*		

## Data Availability

Not applicable.

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
