# Peer review of "Phytotherapy: A Solution to Decrease Antifungal Resistance in the Dental Field"

_biomolecules, 2022, doi:10.3390/biom12060789_

Round 1

Reviewer 1 Report

The authors used  bibliographic searches to investigate and describe phytotherapy resources as a possible solution to oral antifungal resistance. Although the work is important, the authors need to perform major editorial changes to the manuscript. As the manuscript is currently, it is very difficult to read. There are several paragraphs that are very short and need to be integrated into other paragraphs or explained in more detail (e.g., Lines 36-41). For example, the authors need to explain in more detail about possible pharmacological treatments in the introduction and why it is important. 

The authors also need to check for errors, for example species names need to be in italics (Line 37).

Reviewer 2 Report

The manuscript written by Cuenca-León et al has a very interesting theme that would be very welcome to the dentistry area of knowledge. The review aims to describe phytotherapy resources as possible solutions or alternative treatments to overcome antifungal resistance to the conventional antifungal drugs in the oral cavity.

However, in this reviewer's opinion, the main subject of the review is not comprehensively described and the paper's goal was not reached.

Importantly, the manuscript is not very well written and extensive editing of the English language and style is mandatory!

Therefore, I recommend this paper be reconsidered after a major revision.

Other Comments:

Do not use capital letters in Tables 1, 2, and 3.

Delete Table 2 and add the mechanism of action of the drugs in Table 1.

What are the exclusion criteria used in the literature search?

The description of the resistance mechanisms of antifungal drugs is not the paper's main aim. Please revise and summarize section 3.4.

Line 198: Correct “Physiotherapeutic” –

In lines 203 and 222 write the name of the authors that performed the study. For example, Uguna et al…

Section 3.5 must be revised and the results should be better described. The authors only described the results from 5 references!

Table 3 is very incomplete. The columns of the table should describe: Plant Family/ Botanical Name/ Common name/ Bioactive compounds/ Yeast/ Major findings

Moreover, what type of plant extracts was used in all of these studies? Aqueous extracts? Ethanol extracts? This is an important type of information that should also be described in the table.

Please revise the main bioactive compounds for all references, because some of them don´t make any sense, such as phenol (reference 30); ethyl acetate (reference 33); mineral acid (reference 35), etc.

Moreover, what were the criteria used to include the described references/studies in table 3? Do all of these studies refer to plant alternatives to overcome drug resistance in the oral cavity? Because it seems to me that most of the described studies are only about some plant extracts that were tested against Candida sp, with no further relationship with the main goal of this paper.

The discussion section must be thoroughly revised and improved; the paper as it is seems only a collection of data and lacks a proper discussion, particularly regarding the phytotherapy alternatives to decrease antifungal resistance in the dental field.  In all of the long discussion section, only 5 paragraphs are related to phytotherapy!  It´s mandatory to make a critical assessment of all the collected data. An improved discussion and conclusion will help the reader to make decisions about future studies directions, and what is already shown to be promising and not promising. Some clear conclusions for future work should also be given.

Round 2

Reviewer 1 Report

The authors have have addressed most of the problems. However, the paragraph structure in the Discussion still needs to be addressed. For example, some paragraphs need to be joined.  Also, there are formatting errors. For example, species names need to be in italics.

Author Response

Estimado revisor, gracias por sus observaciones y comentarios sobre el artículo enviado.

Sus notas nos han permitido mejorar las deficiencias y corregir algunos errores detectados en el texto.

Reviewer 2 Report

The authors have addressed some of the previously reported problems. However, all the text must be reviewed and corrected because some parts are not clear.

For example,  table 2 is incomplete. The main findings for all the reported studies are not described; what is the reason to have a column with the main findings if nothing is written?

Botanical names of the plant species must be in italics - revise all the text, including the table

Section 3.5 - Correct Phytotherapic. The correct term is Phytotherapy

Section 3.5 should begin by describing the importance of phytotherapy as an alternative to overcome drug resistance. Don´t begin a section describing what is stated by other authors.

Lines 665 - 671 - Please confirm this study. There's no sense of what is described in these lines.

Lines 678- 686 - Please revise and re-write this paragraph.

All the manuscript MUST be carefully read and revised by the authors to improve the clarity and coherence of the text. After that, another English revision will be mandatory.

Author Response

Dear reviewer, thank you for your observations and comments on the article submitted.

Your notes have allowed us to improve the deficiencies and correct some errors detected in the text.
